

# 1    Climate change as a driver of future human migration

Min Chen[1,2] and Ken Caldeira[1*]
[1]Department of Global Ecology, Carnegie Institution for Science. 260 Panama Street, Stanford,
CA 94305, USA
[2]Joint Global Change Research Institute, Pacific Northwest National Laboratory, 5825 University
Research Court, Suite 3500, College Park, MD 20740, USA
**Corresponding author**
Ken Caldeira (kcaldeira@carnegiescience.edu)





**Abstract**
Human migration is both motivated and constrained by a multitude of socioeconomic and
environmental factors, including climate-related factors. Climatic factors exert an influence on
local and regional population density. Here, we examine implications for future motivation for
humans to migrate by analyzing today's relationships between climatic factors and population
density, with all other factors held constant. Such 'all other factors held constant' analyses are
unlikely to make quantitatively accurate predictions but the order-of-magnitude and spatial pattern
that come out of such an analysis can help inform discussions about the influence of climate change
on the possible scale and pattern of future incentives to migrate. Our results indicate that, within
decades, climate change may provide to hundreds of millions of people additional incentive to
migrate, largely from warm tropical and subtropical countries to cooler temperate countries, with
India being the country with the greatest number of people with additional incentive to migrate.
These climate-driven incentives would be among the broader constellation of incentives that
influence migration decisions. Areas with the highest projected population growth rates tend to be
areas that are likely to be most adversely affected by climate change.



## 1. Introduction


Human migration is a complex socioeconomic phenomena driven by mixture of historical, political,
cultural, economic and geographical factors (Greenwood 1985), often by the need to adapt to
environmental stressors (Adger et al. 2014) including those caused by climate change (Myers 1993;
Núñez et al. 2002; Stapleton et al. 2017; Missirian and Schlenker 2017). Climate change is
expected to lead to higher temperatures and an altered hydrological cycle in the coming decades
(McLeman and Hunter 2010), and temperature and precipitation changes have been shown to
influence human migration at local to regional scale (Barrios et al. 2006; Black et al. 2011;
Marchiori et al. 2012; Gray and Bilsborrow 2013; Hsiang et al. 2013; Mueller et al. 2014; Bohra-
Mishra et al. 2014; Kelley et al. 2015).
We apply a simple and transparent approach to estimate the number and geographic distribution
of people for whom temperature and precipitation changes may provide an additional incentive
migrate. Of course, people are subject to a wide range of incentives and constraints; therefore,
actual future migration will depend on a much broader set of factors (Greenwood 1985; Adger et
al. 2014). Ideally, projections of future human migration patterns would involve consideration of
a wide range difficult-to-quantify factors (e.g., future wealth, efficacy of adaptive response,
cultural factors, and non-linear interactions between climate change and population growth)
(Holobinko 2012; Suweis 2018). Our goal is to identify what continuance of current relationships
between climate variables and human population density would imply for future incentives to
migrate. While these relationships will not remain fixed in time, it is nonetheless useful to
understand what direct application of current relationships to future climate would contribute to
the set of incentives that will influence future human migration.



## 2. Methods


2.1 Overview
Nordhaus (2006) applied a regression analysis on geographic and economic data to estimate the
influence of climate variables on the areal density of Gross Domestic Product (GDP). Samson et
al. (2011) used weighted regression model to identify ideal temperature and precipitation ranges
for human habitation (as measured by population density), and studied how those ideal temperature
and precipitation ranges may change in the future owing to climate change. Here we apply similar
methods to the same dataset, the Geographically based Economic data (G-Econ), to estimate the
influence of climate variables on population density.
To estimate of the influence of climate on the attractiveness of different locations, we apply the
historical relationship between climate variables and population density, along with projections
(Taylor et al. 2012) of future climate change from the output of the Coupled Model
Intercomparison Project Phase 5 (CMIP5) under Representative Concentration Pathways (Vuuren
et al. 2011) (RCPs, including RCP 2.6, RCP 4.5, RCP 6.0 and RCP 8.5) scenarios, incorporating
future country-scale demographic population projections   from the United Nations' World
Population Prospects 2015 (United Nations, 2015). Details are in the Analysis section below, but
the basic idea is that if, for example, historical relationships between population density and
climate change would predict a 10% decrease in population density for a grid cell in a climate
change scenario, we would estimate that there would be incentive for 10% of the future population
(as estimated by the UN) to migrate from that grid cell. Of course, many other factors including
family ties, linguistic barriers, lack of resources, employments relations, and so on, would be
expected to influence migration decisions.




2.2 Data
This research uses the Geographically based Economic data (G-Econ) dataset (Nordhaus 2006) for
the historical climate and population data. The G-Econ dataset is originally developed for
analyzing global economic activities and provides gridded (1°×1°) economic (e.g. Gross Cell
Product, population) and geographical (e.g., climate, location, country, distance from seacoasts,
soils and vegetation cover) information covering all terrestrial regions. In total, there are 27,445
grid cells in the dataset. G-Econ's climatology data, including annual mean air temperature ($T$,
in °C) and annual precipitation ($P$, in mm year$^{-1}$), were derived from the Climate Research Unit
Average Climatology high-resolution data sets (New et al. 2002). The gridded population ($N$) was
adapted      from      the      Gridded      Population      of      the      World      (GPW)      dataset
(http://sedac.ciesin.columbia.edu/data/collection/gpw-v3).   More details and the data download
link is available at http://gecon.yale.edu/.
In this study, from the G-Econ dataset, we used the population density ($D$) and the geographical
data, including $T$, $P$, distance to lake ($DL$, in $km$), distance to major river ($DMR$, in $km$), distance
to river ($DR$, in $km$), distance to ocean ($DO$, in $km$), elevation ($E$, in $m$), and surface roughness
(*Roughness*, in $m$).
To make our projections, we used $T$ and $P$ in historical (*i.e.*, 1960-2005) climate, and future climate
scenarios (2006-2100) from the output of the Coupled Model Intercomparison Project Phase 5
(CMIP5), which produces state-of-art multi-model dataset to advance the knowledge of climate
change. We collected the model projected $T$ and $P$ (20 model projects; see Table A1) under all
Representative Concentration Pathways (RCPs, including RCP 2.6, RCP 4.5, RCP 6.0 and RCP



8.5) from CMIP5 dataset to represent the range of future climate projections. We regridded the
CMIP5 data to a 1°×1° common grid using bilinear interpolation.
We used the historical and predicted (median-variant) country-level population data from the
World Population Prospects: The 2015 Revision by the United Nations Department of Economic
and Social Affairs (United Nations, 2015). We use $W_{i,y}$ to denote the population estimated by the
UN for grid cell $i$ in year $y$; we use $W_{c,y}$ to denote the population estimated by the UN for country
$c$ in year $y$.
2.3 Analysis
*Year 2005 population density and within-country distribution.* Areal population density for year
2005 in each grid cell $i$ ($D_i$) was calculated from the population ($N_i$) of 2005, grid area ($A_i$, in km$^2$)
and land fraction of the grid ($L_i$, no unit) from G-Econ dataset:

$$D_i = N_i \big/ \left( A_i \times L_i \right) \tag{1}$$

We denote the fraction of population of country $c$ living in grid cell $i$ with the symbol $d_{i,c}$:

$$d_i = N_i \big/ \sum_{i \in c} N_i \tag{2}$$

where $i \in c$ indicates that the summation is performed over all grid cells in country $c$. The
distributional parameter, $d_{i,c}$, is considered to be constant in time.
*Linear regression model.* Our methods for estimating climate influence on population density
parallels methods previously applied (Nordhaus 2006) to estimate climate influence on areal
density of GDP. The basic idea is to find a single set of coefficients that explain within-country
relationships between population, climatic and geographic variables. For our regressions, we used



data from the G-Econ dataset[18] and the Climate Research Unit Average Climatology high-
resolution data sets[27] (for filling the missing data in the G-Econ dataset). To estimate logarithm of
population density from both geographical (**G**) and climatic variables (**C**), we used the equation:
$$\log_{10} D = \beta_0 + \mathbf{G}\boldsymbol{\beta_G} + \mathbf{C}\boldsymbol{\beta_C} \tag{3}$$
where $D$ is a vector of grid-scale population densities (i.e., $D_i$ for grid cell $i$). Specifically,
$$\mathbf{G} = \begin{bmatrix} country & soil & DL & DMR & DR & DO & E & roughness \end{bmatrix} \tag{4}$$
$$\mathbf{C} = \begin{bmatrix} T & T^2 & T^3 & p & p^2 & p^3 & Tp & T^2p & p^2T \end{bmatrix} \tag{5}$$
where $T$ is as defined above, and $p$ is $\log_{10} P$. *country* and *soil* are categorical variables, $\boldsymbol{\beta_G}$ and $\boldsymbol{\beta_C}$
are numerical coefficients vector on geographical and climatic variables, respectively.
$$\boldsymbol{\beta_G} = \mathrm{Transpose}\begin{bmatrix} \beta_{G,country} & \beta_{G,soil} & \beta_{G,DL} & \beta_{G,DMR} & \beta_{G,DR} & \beta_{G,DO} & \beta_{G,E} & \beta_{G,roughness} \end{bmatrix} \tag{6}$$
and
$$\boldsymbol{\beta_C} = \mathrm{Transpose}\begin{bmatrix} \beta_{C,T} & \beta_{C,T^2} & \beta_{C,T^3} & \beta_{C,p} & \beta_{C,p^2} & \beta_{C,p^3} & \beta_{C,Tp} & \beta_{C,T^2p} & \beta_{C,p^2T} \end{bmatrix} \tag{7}$$
Antarctica, Greenland, and grid cells with zero precipitation were excluded from this analysis.
The values for the $\beta$-coefficients are determined by an area-weighted ordinary-least-squares curve
fit to $\log_{10}$ D. Fitting the above linear regression model was conducted in MATLAB R2017a
(http://www.mathworks.com/products/matlab/). In total, 20,503 grid cells had data for all
parameters needed for the fitting procedure. Variability that is not explained by equation (3) is
assumed to be the result of unknown factors which we treat as invariant with time.



*Population change projections.* We first calculated the ratio of population in the changed climate
relative to the base-state climate (here taken to be the climate in the period preceding 2005) in
region $i$ for the climate in year $y$ considering climate factors alone ($r_{i,y}$):

$$r_{i,y} = \frac{D_{i,y}}{D_{i,2005}} \tag{8}$$

For each grid, we calculated $r_{i,y}$ for each year from 2006 to 2100 using equation (8) and 30-year
moving average of $T$ and $P$ projected by each CMIP5 model. (The 30-year moving average ends
on the period under consideration so that decisions are made on past but not future climate states.)
In the absence of climate change, we would estimate the population in grid cell $i$ in country $c$ for
year $y$ ($W_{i,y}$) to be $d_{i,c} \times W_{c,y}$, where $c$ is the country containing grid cell $i$. If we directly apply the
population change ratio under climate change ($r_{i,y}$) to the population estimates, the population with
taking climate change into account would be $r_{i,y} \times W_{i,y}$. However, this estimate must be scaled to
conserve total population. Thus, the population $N_{i,y}$ of grid cell $i$ in year $y$ can be estimated to be:

$$N_{i,y} = r_{i,y} \times W_{i,y} \times \frac{\sum_{i \in c} d_{i,c} \times W_{c,y}}{\sum_{i \in c} r_{i,y} \times d_{i,c} \times W_{c,y}} \tag{9}$$

By doing this adjustment, we conserve the world total population, but take climate change into
account to estimate the spatial distribution of population.
We then estimate the number of people for whom climate change is projected to provide additional
incentive to migrate for grid-cell $i$ and year $y$ (indicated by $\Delta N_{i,y}$) as:

$$\Delta N_{i,y} = N_{i,y} - W_{i,y} \tag{10}$$



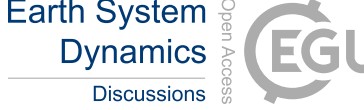

Negative values of $\Delta N_i$, are interpreted as indicating areas where climate change provides
additional incentive to emigrate; positive values indicate areas that are projected to increase in
relative attractiveness. (Even if everyplace were to decrease in absolute attractiveness due to
climate change, the places with a smaller absolute decrease would increase in relative
attractiveness.)
We define $f_{i,y} = N_{i,y} / W_{i,y}$, so that $f_{i,y} - 1$ indicates the fractional change in population that would
be required to offset the influence of climate change on the attractiveness of grid cell $i$ in year $y$.
When $f_{i,y} - 1 < 0$, that means that grid cell $i$ has become less attractive. We integrated $N_{i,y}$ for grid
cells in each country $c$ to yield $N_{c,y}$ and define $f_{c,y} = N_{c,y} / W_{c,y}$. We calculate results independently
for each of the CMIP5 models simulations (Taylor et al. 2012) and present median results.
Where a range is reported, it encompasses results for 68% of the CMIP5 models.
We report results with two significant digits. The computer scripts written in Matlab R2017a used
to perform our analyses are available upon request.

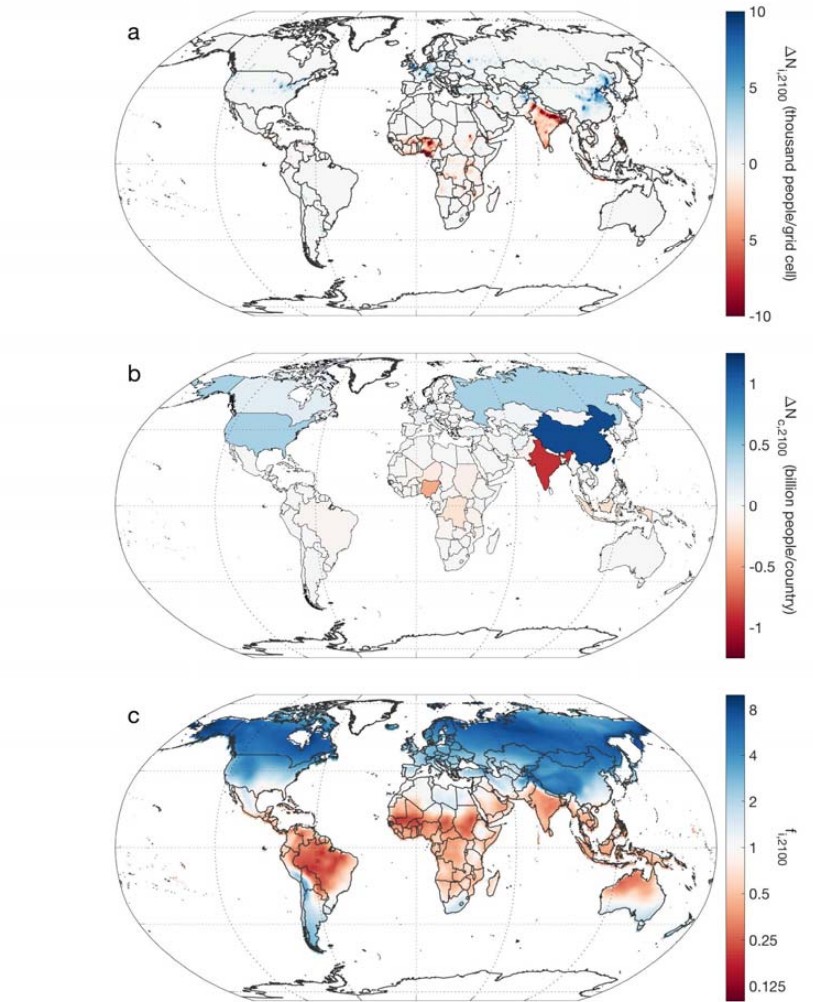

Figure 1. The number of people for whom climate change is projected to provide additional incentive to migrate under RCP 8.5 per $1° \times 1°$ grid cell ($\Delta N_{i,2100}$, in thousand people, panel a) and per country ($\Delta N_{c,2100}$, in billion people, panel b). The fractional change in population that would be required to offset the influence of climate change on the relative attractiveness of living in a particular location for year 2100 ($f_{i,2100}$) under scenario RCP 8.5 (c). To isolate the effect of climate change on incentives to migrate, all factors are held constant, except for climate and country-level population. Of course, many other factors influence migration decisions.



## Results

The regression of population density against geographic and climate variables as described above (see also Methods and Supporting Material) explains 72% of the geographic variance in the logarithm of population density. Applying our regression equation to climate model and demographic projections, we find that $\Delta N_{i,y}$ is negative (i.e., indicating decreased attractiveness) in regions that are already hot and are projected to experience substantial additional warming under climate change (primarily tropical and subtropical regions), whereas we find that $\Delta N_{i,y}$ is positive (i.e., indicating increased attractiveness) in cooler regions (primarily in the temperate regions of the Northern Hemisphere; Figure 1a and A1,a,b,c).

Under RCP 8.5, India has the largest negative $\Delta N_{c,2100}$ value among countries (0.89 [0.77 to 1.10] billion; Figure 1b), followed by Nigeria (0.46 [0.38 to 0.58] billion). The other countries with the largest negative values of $\Delta N_{i,2100}$ are Democratic Republic of Congo (0.20 billion), Indonesia (0.18 billion), Niger (0.14 billion), Sudan (0.11 billion), Philippines (0.10 billion), Bangladesh (0.09 billion), Tanzania (0.09 billion) and Pakistan (0.08 billion). In contrast, China, Russia and the United States all have positive values of $\Delta N_{c,2100}$.

The metric $f_{i,2100}$ is less than 0.3 in parts of the Northern African Tropical Savanna, Tropical South America and Tropical Asia under RCP 8.5, indicating that future incentives to migrate from those areas may be substantial. The metric $f_{i,2100}$ is >5 in much of Canada, Russia and Scandinavia, and parts of the United States, and China (Figure 1c), which could indicate that in the absence of other barriers these regions could become migration destinations. Results for RCP 2.6, 4.5 and 6.0 show similar spatial patterns but at lower magnitude (Figure A1).



The countries with the largest projected population growth to year 2100 tend to be countries where
the largest negative values of $\Delta N_{c,2100}$ (Fig. 2). The equation $\Delta N_{c,2100} = (1.79 \pm 0.06)\, \Delta W_{c,2100} +$
$(0.21 \pm 0.02)$ explains 79% of the variation in population-weighted $\Delta N_{c,2100}$ (best estimate $\pm$ 1
standard error). Figure 2 shows average projected population increase from 2005 to 2100 ($\Delta W_{c,2100}$)
on the horizontal axis is negatively correlated to the number of people in each country with
additional incentive to emigrate ($\Delta N_{c,2100}$) on the vertical axis. About 70% of the world's projected
year 2100 population lives in a country that is expected to experience population growth and for
which $\Delta N_{c,2100}$ is < 0 (lower right quadrant in Fig. 2). In contrast, 14% of the global population in
2100 is projected to live in a country experiencing with a population lower than today and for
which $\Delta N_{c,2100}$ is > 0 (upper left quadrant in Fig. 2). Similar patterns are found under other
scenarios (Fig. A2).
Figures 3 shows values of $\Delta N_{i,y}$ integrated over all grid cells with $\Delta N_{i,y} < 0$, indicating the number
of people for whom climate change for whom climate change may produce an additional incentive
to migrate. Under all of the RCP scenarios, this integrated value increases over the next few
decades (Figure 3), reaching 0.6 to 1.9 billion people by 2050 (depending on RCP scenario). By
year 2100 under RCP 8.5, this number increases to about 3.8 [3.3 to 4.9] billion people, which is
about one-third of the projected global population in 2100.





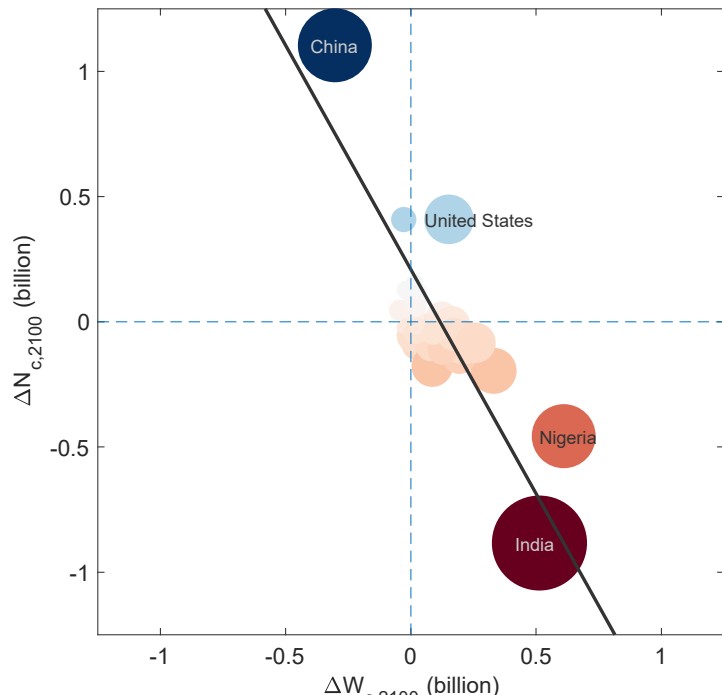


Figure 2. Country-level projections for population increase in year 2100 relative to year 2005 ($\Delta W_{c,2100}=W_{c,2100}-W_{c,2005}$, horizontal axis) and the number of people for whom climate change is projected to provide additional incentive to migrate under RCP 8.5 ($\Delta N_{c,2100}$; vertical axis). Areas of circles are proportional to year 2100 population. Color scale is as per Figure 1b. The line shows the population-weighted linear trend. Negative values on the vertical axis indicate additional incentive to emigrate; positive values indicate countries that increase in relative attractiveness. Results hold all factors constant, except for climate and country-level population.

219



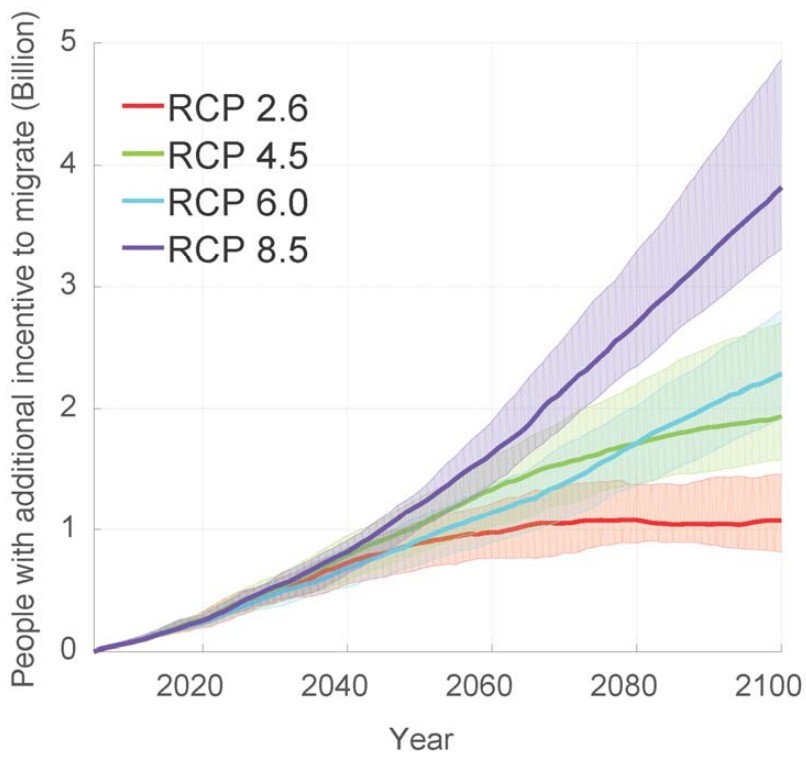

220

Figure 3. Number of people projected to experience additional climate-related incentive to emigrate under
four Representative Concentration Pathways. The lines show the median value across CMIP5 models with
results from 66 % of the models falling within the shaded area. Results hold all factors constant, except for
climate and country-level population.




## Discussion and Conclusions

In this section, we discuss some of the relevance of the results of our calculations for the real world.
We intend our quantitative results to indicate possible orders-of-magnitude and global-scale spatial
patterns of people with changed incentives; we do not intend our results to be interpreted as
quantitative predictions of future climate-induced human migration.
Our calculations take into account changes in temperature and precipitation only, under the
artificial assumption that all other factors remain constant. Our highly idealized calculations are
intended to indicate the scale and geographic distribution of people for whom climate change
might provide an additional incentive to migrate. Our calculations also indicate which regions
climate change might make more attractive to potential migrants. Clearly, migration decisions are
influenced by a wide range of factors (McLeman and Hunter 2010; Fussell et al. 2014). Further,
there is often a substantial incentive to avoid migration entirely, so additional incentive to migrate
does not imply an overall positive net incentive to migrate. The number of people who will have
positive net incentive to migrate as a result of climate change is thus less than the number of people
for whom climate change will provide an additional incentive to migrate. Migration is one of many
possible adaptive responses to climate change. For example, people might choose to cool interior
spaces with air conditioners (Barreca et al. 2016). Another response could be to shift from
agricultural work in rural environments to industrial or service-sector jobs in more urbanized
environments (Neill et al. 2010; Jiang and O'Neill 2017), and thus migration flows can be
influenced by differences in types of development and not only climatic factors.
Our results indicate that India may be the country that will contain the largest number of people to
whom climate change may provide an additional incentive to emigrate. West Africa, and in
particular, Nigeria, may be the second most important area in this regard (Figure 1a,b). This is





largely a consequence of high population densities in areas that are already warm and projected to
get warmer. Our results indicate that many people living in the Amazon region would have
additional incentive to emigrate, but population density is generally low. More generally, climate
change may provide additional incentive to emigrate to many people living in the tropics (Figure
1c). In contrast, our regression equations indicate that, from a purely climatic perspective, climate
change may increase the attractiveness of northern countries, such as China, Russia, Canada,
Norway, Sweden and Finland, relative to most other parts of the world.
There is a country-level correlation between projected population increase and the degree to which
climate change is projected to provide an additional incentive to emigrate. This correlation
suggests that population increases have the potential for exacerbating negative effects of climate
change in much of the world. Over two-thirds of the world's year 2100 population is projected to
live in a country with greater population than today and for which climate change may provide
additional incentive to emigrate. In contrast, about one out of seven people are projected to live in
a country with a lower population and where climate change may cause to become relatively more
attractive. China is the largest country that is expected to both experience a decrease in population
and an increase in climate-related relative attractiveness. Moreover, our calculations suggest that
India could be the largest potential source of climate emigrants, and that China could potentially
be the largest potential destination for climate immigrants (Figure 1b). However, immigration in
China is currently very limited (Abel and Sander 2014). Thus, barriers to migration in southeast
Asia could potentially become an important source of future climate-related conflict (Hsiang et al.

2013).

Climate change may provide additional incentive to migrate to hundreds of millions of people
within the next decades and potentially billions of people by the end of this century (Figure 3).



The number of people projected to have additional incentive to migrate by year 2100 under RCP
4.5 or 6.0 is about half that projected under RCP 8.5, and the number project under RCP 2.6 is
about half that projected under RCP 4.5 or 6.0. This result points to the important role that
emissions reductions may play in reducing climate-related incentives to migrate. Successful local
adaptation measures could greatly reduce incentives to migrate (Adger et al. 2014).
Climate change is likely to induce a complex web of dynamical interactions at a range of spatial
and temporal scales, and these interactions are not well represented by our model. For example,
considerations of language, work, and family ties can provide strong incentive not to migrate.
Projections of how climate change might affect migration are therefore fraught with uncertainty.
Nevertheless, the results of our calculations may indicate areas that climate change can be expected
provide large numbers of people, primarily in the tropics, an additional incentive to migrate,
primarily to the middle and high latitudes of the Northern Hemisphere. This change in climate-
driven incentives to migrate is one factor among many that need to be included in a comprehensive
understanding of possible future migration flows.

**Code/Data availability**
All the data used in this study is publicly available. The CMIP5 climate projections are available
at      https://cmip.llnl.gov/cmip5/data_portal.html.      The      G-Econ      dataset      is      available      at
http://gecon.yale.edu/. The WPP2015 (World Population Prospects: The 2015 Revision by the
United    Nations    Department    of    Economic    and    Social    Affairs)    data    is    available    at
http://esa.un.org/unpd/wpp/Download/Standard/Population/.



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

**Acknowledgements**
The authors thank Bill Hayes for his efforts on processing CMIP5 data. We appreciate comments
from Kate Ricke and Juan Moreno-Cruz on earlier drafts of this manuscript. This work supported
by the Carnegie Institution for Science endowment and the Fund for Innovative Climate and
Energy Research.

**Author Contributions**
M. C. and K. C. conceived and designed the project and performed the computational analysis.
M.C. wrote the first draft of the manuscript with later development from K. C.

**Competing interests**
The author(s) declare no competing interests.



Table A1. CMIP5 models used in this study.

| Model | Country and Research Center | Resolution (Latitude, Longitude) |
|---|---|---|
| CCSM4 | United States, NCAR | (0.9424, 1.25) |
| CESM1-CAM5 | United States, NCAR | (0.9424, 1.25) |
| CSIRO-Mk3.6.0 | Australia, CSIRO | (1.8653, 1.875) |
| FIO-ESM | China, The First Institute of Oceanography, SOA | (2.8125, 2.8125) |
| GFDL-CM3 | United States, NOAA/GFDL | (2, 2.5) |
| GFDL-ESM2G | United States, NOAA/GFDL | (2.0225, 2) |
| GFDL-ESM2M | United States, NOAA/GFDL | (2.0225, 2.5) |
| GISS-E2-H | United States, NASA GISS | (2, 2.5) |
| GISS-E2-R | United States, NASA GISS | (2, 2.5) |
| HadGEM2-AO | United Kingdom, MOHC | (1.25, 1.875) |
| IPSL-CM5A-LR | France, IPSL | (1.8947, 3.75) |
| IPSL-CM5A-MR | France, IPSL | (1.2676, 2.5) |
| MIROC-ESM | Japan, JAMSTEC; Atmosphere and Ocean Research Institute (AORI); National Institute for Environmental Studies (NIES) | (2.7906, 2.8125) |
| MIROC-ESM-CHEM | Japan, JAMSTEC; AORI; NIES | (2.7906, 2.8125) |
| MIROC5 | Japan, JAMSTEC; AORI; NIES | (1.4008, 1.40625) |
| MRI-CGCM3 | Japan, MRI | (1.12148, 1.125) |
| NorESM1-M | Norway, Norwegian Climate Centre | (1.8947, 2.5) |
| NorESM1-ME | Norway, Norwegian Climate Centre | (1.8947, 2.5) |
| BCC-CSM1.1 | China, BCC | (2.8125, 2.8125) |
| BCC-CSM1.1-M | China, BCC | (1.125, 1.125) |

383



Figure A1. The number of people for whom climate change is projected to provide additional incentive to migrate under RCP 2.6, 4.5 and 6.0 per $1° \times 1°$ grid cell ($\Delta N_{i,2100}$, in thousand people) and per country ($\Delta N_{c,2100}$, in billion people). The fractional change in population that would be required to offset the influence of climate change on the relative attractiveness of living in a particular location for year 2100 ($f_{i,2100}$) under the scenarios. The three rows presents $\Delta N_{i,2100}$, $\Delta N_{c,2100}$ and $f_{i,2100}$ under RCP 2.6, 4.5 and 6.0 (columns), respectively. Color schemes are the same as in Fig. 1. Results hold all factors constant, except for climate and country-level population.

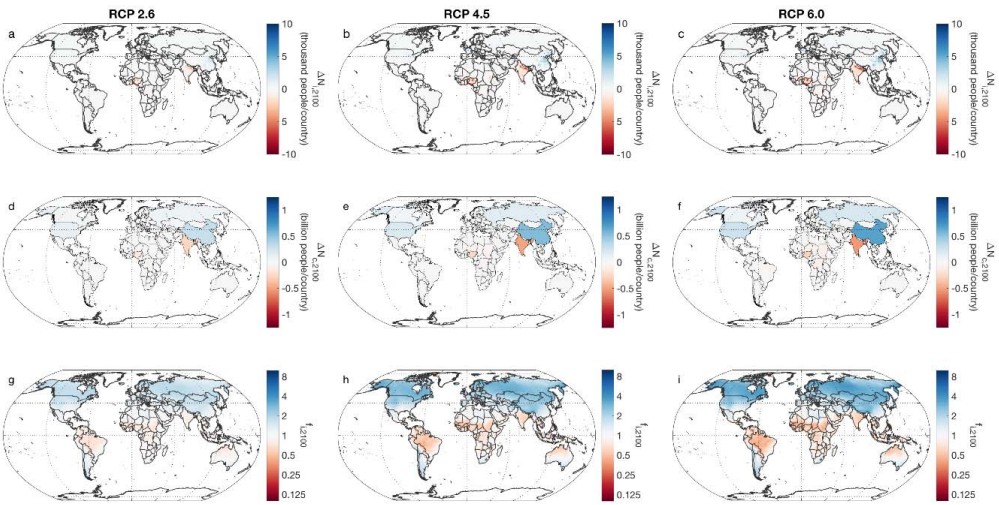

391

392





Figure A2. Country-level projections for population increase in year 2100 relative to year 2005 ($\Delta W_{c,2100}=W_{c,2100} - W_{c,2005}$, horizontal axis) and the number of people for whom climate change is projected to provide additional incentive to migrate under RCP 2.6, 4.5 and 6.0 ($\Delta N_{c,2100}$; vertical axis). Areas of circles are proportional to year 2100 population. Color scale is as per Figure 2. The line shows the population-weighted linear trend by fitting $\Delta N_{c,2100}=a\Delta W_{c,2100}+b$, where $a$ and $b$ are parameters. For RCP 2.6, $a=-0.49\pm0.06$, $b=0.06\pm0.02$ (best estimate $\pm$ 1 standard error), and $R^2=0.80$; for RCP 4.5, $a=-0.92\pm0.06$, $b=0.10\pm0.02$, and $R^2=0.79$; for RCP 6.0, $a=-1.08\pm0.06$, $b=0.13\pm0.02$, and $R^2=0.79$. Negative values on the vertical axis indicate additional incentive to emigrate; positive values indicate countries that increase in relative attractiveness. Results hold all factors constant, except for climate and country-level population.

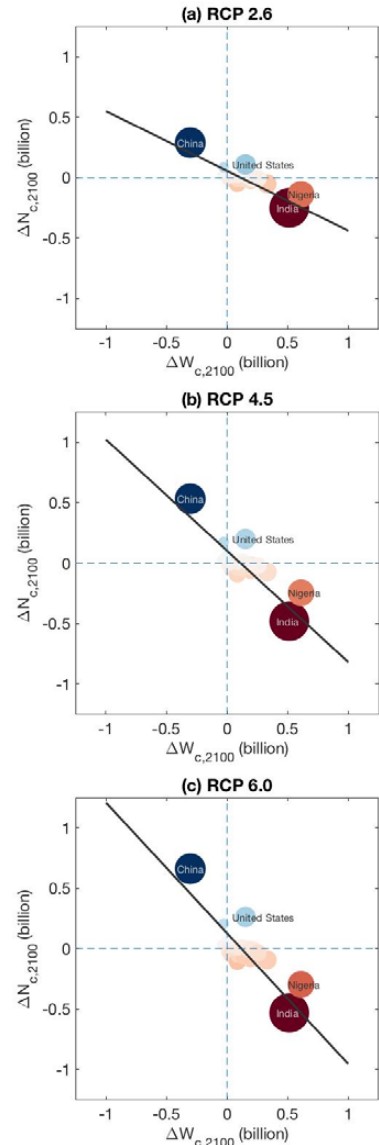