# Peer review of "Climate change as a driver of future human migration"

_Earth System Dynamics, 2019_

## Short Comment (SC1) · 11 Feb 2020

Two comments on your article: 1) rather than presenting the results in terms of RCPs, present them in terms of per degree of global warming. This should simplify your results. GCMs have robust responses in precipitation and temperature (Fischer et al., 2014), and heat stress (Buzan and Huber, 2020, in press).

2) The covariance of temperature-humidity-radiation is overlooked in your analysis, and likely underestimates the impact of unmitigated heat stress—something that T alone cannot resolve. These moist heat stress conditions can vary over the tropics. For example, at 3C of global warming, South East Asia experiences larger reductions in labor capacity (40-60% total annual capacity) as compared to India (50-80% total

annual capacity). If you calculate this covariance, you will need sub daily variables, as monthly data creates errors larger than the climate change signal (Buzan and Huber, 2020, in press).

Buzan and Huber, 2020: Moist heat stress on a hotter Earth, Annual Review of Earth and Planetary Sciences, in press.

Fischer et al., 2014: Models agree on forced response pattern of precipitation and temperature extremes, GRL, https://doi.org/10.1002/2014GL062018.

---

## Short Comment (SC2) · 11 Feb 2020

Methods section: Consider adding a table that summarizes geographical data variables including the rationale for using them in this analysis.

Figures: Figure 1b is redundant with vertical axis of Figure 2. I suggest removing panel b from Figure 1 and panels d-f from Figure A1 and combining panels 1a, 1c, A1a-c, and A1g-i into a revised Figure 1.

I suggest combining Figure 2 and Figure A2 into a four panel Figure 2. To improve clarity of Figure 2, replace country names with numbers in circles and add number/country identification in the legend. Regarding Figure 2 legend, add a, b, and Rˆ2 for RCP8.5 linear trend.

---

## Referee Comment (RC1) · Anonymous Referee #1 · 3 May 2020

The paper gives a rough estimate of the incentive of people to move from their country of origin to another country with increasing climate change using an innovative approach. The paper is largely technical: It starts from estimating a plausible black-box relationship between population density and climate variables. The authors than proceed to project differences between population densities expected by standard UN population estimates and population densities estimated by combining climate change scenarios with their estimated relationship between climate variables and population density. Differences are than interpreted to provide incentives for people to move. The authors make clear throughout this paper, that they are not providing an actual estimate of future population movements due to climate change, pointing to the variety of factors for migration which they do not consider. They are obviously aware of the

controversial nature of projections of migration related to climate change, but it would be good if this insight could be strengthened by a few references to the broader social science literature on migration and climate change, which emphasis the lack of certainty of projections, for instance the recent letter from a number of researcher printed in Nature Climate Change (Boas, I., Farbotko, C., Adams, H. et al. Climate migration myths. Nat. Clim. Chang. 9, 901–903 (2019).) I also recommend adding a bit more on why the authors think that the correlation between grid cell population density and climate variables is not by coincidence. They reference a bit of literature on this, but some substantial arguments would be appreciated. A brief discussion of the mechanisms at work would also help the reader to be more aware of which of the factors held constant in the authors' analysis would like be of major importance in shaping the future relationship between population densities and climate variables. This might also lead the authors to reconsider their title, which, while not wrong, might lead to the misunderstanding that they are making predictions, when they are clear in the text that they do not. The exposition of approach and methodology is generally clear and convincing. On the technical side (some of these questions may be addressed in the supplementary material which was not available to me) I have the following questions and suggestions: • The authors need to juggle between grid cell and country level data, because of data availability. It would be useful if they briefly discussed the implications and potential problems of this. • Related to this: If I understand correctly, the authors need to keep the within-country distribution of population constant both for the projection periods, implying that all migration is international, while, in reality, some of the incentive for migration may be internal. • It would be good to know about which of the variables had what importance etc. I missed a table with results for the regression between population density and climate variables. • I have a hunch that the relationship between population density and climate variables is not constant over time but trending. This would require some adjustment in estimation as well as influence projection. • Related to this: the time period for estimating the relationship is considerably shorter than that for the projection. Potential implications should at least

be mentioned. • I checked the UN Population data and it looks to me as if there are various projections, with and without migration figured it. Which one did the authors use?

---

## Referee Comment (RC2) · Anonymous Referee #2 · 20 May 2020

This manuscript examines aims at examining the effects of future climatic changes on people's incentives to migrate. Starting with econometric results based on historical (i.e., 1960-2005) climatic (temperature and precipitation), population density, and geographical data, the authors make projections of the climatic variables under the four Representative Concentration Pathways (RCPs) on future (i.e., 2006-2100) population density. They report that climate change may soon force hundreds of millions of people to migrate mostly from warm tropical and subtropical countries to cooler temperate countries (i.e., regions of the Northern Hemisphere). In addition, individual countries such as India, Nigeria, and Democratic Republic of Congo are predicted to have the greatest number of people with additional incentives to migrate, while China, Russia, and the USA are expected to exhibit greater attractiveness and hence become migration destinations. The manuscript, without any doubt, studies an interesting and very important question in the academic and policy worlds. It also shows a methodological sophistication and rigor, but provides evidence in support of propositions, which are already known (see for example the IPCC 5th AR (2014) and the 1.50C Report (2018). Unfortunately, I believe that this manuscript does not advance our understanding of the climate-migration relationship and as a result, it does not make any significant contribution to the relevant literature. Hence, I simply cannot recommend publication of this manuscript. There are three main reasons: 1. Although the article cites some literature on the relationship between climate change and migration (alas, a relatively old one that focuses only on a positive relationship), it does not really engage with this literature. As such, no attempt is made to explain what we know and what we don't know. This makes it impossible for the reader to see what new knowledge this paper is bringing to the table. As far as I can see, this paper does not really advance on the state of knowledge on climate change and migration. 2. The authors do acknowledge that migration is inherently multiclausal, i.e., migration decisions depend on economic, social, demographic, political, and environmental/climatic factors. However, at the end they chose to examine the climate-migration nexus in a direct and consequently deterministic manner, and make big and bold claims, which have huge implications of how the social consequences of climate change could be dealt with, based on false and naive assumptions. It can still be possible to examine within this multi-causal context what the impact of the climate has been and to examine its influence – but to fully categorize it as something separate, and to base predictions, numbers and conclusions on that, is overly simplistic. A good start is to read the 2010 Foresight report on environmental change and human migration and resulting publications from Black et al (2011). For a good overview of the debate see Piguet (2013) and McLeman/Gemenne (2018). 3. The manuscript also disregards many years of similar research: for instance, Hsiang and Sobel (2016) show that tropical populations would have to travel distances greater than 1000 km over less than a century if global mean temperature rises by 2 °C over the same period (they also present a figure very similar to Figure 1 the authors present

in the manuscript, albeit the authors use different RCPs). Similarly, a 2018 World Bank study predicts that climatic changes, in particular drought, may cause as many as 143 million people to be displaced within Latin America, Sub-Saharan Africa, and South Asia by 2050 (Kumari Rigaud et al. 2018). Hence, I can't see how this manuscript brings anything new to the table.

Black, Richard; W Neil Adger, Nigel W Arnell, Stefan Dercon, Andrew Geddes & David SG Thomas (2011) The effect of environmental change on human migration. Global Environmental Change 21: 3-11. Foresight Migration and Global Environmental Change. Final Project Report. London: The Government Office for Science, (2011). Available at: http://www.bis.gov.uk/foresight/ migration Hsiang, S.M. and A.H. Sobel (2016) Potentially extreme population displacement and concentration in the Tropics under non-extreme warming. Scientific Reports 6: 25697. McLeman, Robert & François Gemenne (2018) Environmental migration research: Evolution and current state of the science. In Robert McLeman & François Gemenne (ed.) Routledge Handbook of Environmental Displacement and Migration. London: Routledge, 3-16. Piguet, Etienne (2013) From "primitive migration" to "climate refugees": The curious fate of the natural environment in migration studies. Annals of the Association of American Geographers 103(1): 148-162. Rigaud, Kanta Kumari; Alex de Sherbinin, Bryan Jones, Jonas Bergmann, Viviane Clement, Kayly Ober, Jacob Schewe, Susana Adamo, Brent McCusker, Silke Heuser & Amelia Midgley (2018) Groundswell: Preparing for Internal Climate Migration, World Bank (https://openknowledge.worldbank.org/handle/10986/29461).
* * *

---

## Author Comment (AC1) · 16 Jul 2020

1. Jonathan Buzan suggested that we present results in terms of "per degree warming". We have taken this advice on board and included a new panel to Figure 3 in terms of temperature. This substantially improved the paper. The results collapse nicely when plotted against global mean temperature rather than time. Thank you for this suggestion.

2. Jonathan also pointed out that we ignored the temperature-humidity relationship and its effect on heat stress. We agree that this would be good to do, and a future study could address this. Our project idea was to use the methods and database that Nordhaus used to estimate climate effects on GDP to estimate climate effects on

population density. Humidity would be a good topic to address in future studies. I am not what the correlation is between humidity and precipitation, but some of the humidity effect might be picked up by the precipitation variable.

---

## Author Comment (AC2) · 16 Jul 2020

This comment was to bring together panels from the main paper with similar panels for other cases from the supporting material. We went back and forth with this: Is it better to present everything together and risk overwhelming people with a large number of panels per figure (for one figure it would be 4 cases by 3 variables to give 12 panels) or better to do just one case in the main paper with the remaining cases in supporting material.

It is our judgment that presenting all of the information in the main paper would cause many readers' eyes to glaze over. Thus, we have decided to present a central case in the main paper and show other cases in the supporting material.

[Figure]
James Smoot also recommended some relabeling of figures. We now include a tables in the supporting material supporting each figure, with the numerical data plotted in the figure. This way if the reader wants to plot things a different way or look up a specific country, they can.

We appreciate his comments but ultimately did not take many of them on board.

---

## Author Comment (AC3) · 16 Jul 2020

A formatted version of this response is attached as a supplement.

Referee responses:

RC1: 'Review', Anonymous Referee #1, 03 May 2020

Review comments were very positive, which we appreciate. The main suggestion was to include some additional references and discussion.

They are obviously aware of the controversial nature of projections of migration related to climate change, but it would be good if this insight could be strengthened by a few references to the broader social science literature on migration and climate change,

which emphasis the lack of certainty of projections, for instance the recent letter from a number of researcher printed in Nature Climate Change (Boas, I., Farbotko, C., Adams, H. et al. Climate migration myths. Nat. Clim. Chang. 9, 901–903 (2019).)

This paper is now cited in the very first sentence of our Introduction:

Human migration is a complex socioeconomic phenomena driven by mixture of historical, political, cultural, economic and geographical factors (Black et al., 2011; Boas et al., 2019; Foresight: Migration and Global Environmental Change, 2011; Greenwood, 1985), often by the need to adapt to environmental stressors (Adger et al., 2014) including those caused by climate change (Missirian and Schlenker, 2017; Myers, 1993; Núñez et al., 2002; Stapleton et al., 2017).

It is also cited in another two sentences in our Introduction:

Of course, people are subject to a wide range of incentives and constraints; therefore, actual future migration will depend on a much broader set of factors (Adger et al., 2014; Boas et al., 2019; Greenwood, 1985). Ideally, projections of future human migration patterns would involve consideration of a wide range difficult-to-quantify factors (e.g., future wealth, efficacy of adaptive response, cultural factors, and non-linear interactions between climate change and population growth) (Boas et al., 2019; Holobinko, 2012; Suweis, 2018).

I also recommend adding a bit more on why the authors think that the correlation between grid cell population density and climate variables is not by coincidence. They reference a bit of literature on this, but some substantial arguments would be appreciated. A brief discussion of the mechanisms at work would also help the reader to be more aware of which of the factors held constant in the authors' analysis would like be of major importance in shaping the future relationship between population densities and climate variables.

We now write in our Discussion section:

Parameter values and their uncertainties are shown in Table S2; p-values on coefficients for all temperature and precipitation related variables based on a Student T-test are <0.0005, indicating that these results are unlikely to have been obtained by chance.

Based on observations of maps like the following, we have also added the following text:

It is clear that population distributions are related to climate variables. Population densities tend be very low both in very hot areas (e.g., Death Valley) and in very cold areas (e.g., Alaska), and relatively high in areas with intermediate temperatures (e.g., New York City). Similarly, population densities tend to be low in very dry areas (e.g., central Australia) and very wet areas (e.g., northern Australia) and relatively high where there is an intermediate amount of precipitation (e.g., Sydney, Australia).

This might also lead the authors to reconsider their title, which, while not wrong, might lead to the misunderstanding that they are making predictions, when they are clear in the text that they do not.

Following the referee's suggestion, we have changed the title from

Climate change as a driver of future human migration

To

Climate change as an incentive for future human migration

The authors need to juggle between grid cell and country level data, because of data availability. It would be useful if they briefly discussed the implications and potential problems of this. If I understand correctly, the authors need to keep the within-country distribution of population constant both for the projection periods, implying that all migration is international, while, in reality, some of the incentive for migration may be internal. All of our analysis is at grid cell level, except for future population projections, which are country level data, downscaled to grid level under current population distributed. Further, when we report country-level incentives to migrate, we integrate

over all grid cells in a country, so if one grid cell would be +100 and another -100 within a grid cell, we would report zero incentive to migrate from this country. We have now added the following text to the end of our Introduction:

When we report country-level results, we integrate across all grid cells within a country and report the net value, so our methodology would not predict incentive to migrate from a country that had some grid cells indicating incentives for out-migration but with other grid cells indicating even greater incentive for in-migration.

It would be good to know about which of the variables had what importance etc. I missed a table with results for the regression between population density and climate variables. We are sorry. For some reason our supporting material was not included in our original submission. This information is now in Table S2.

I have a hunch that the relationship between population density and climate variables is not constant over time but trending. This would require some adjustment in estimation as well as influence projection. Related to this: the time period for estimating the relationship is considerably shorter than that for the projection. Potential implications should at least tbe mentioned.

We have now added the following text to the Discussion section.

Further, our calculation treats the relationship between climate and incentive to migrate as constant in time. However, factors such as availability of indoor work in air-conditioned environments would surely modify these relationships. This study isolates a narrow range of factors under ceteris paribus assumptions. We hope our study motivates efforts to quantitatively address the panoply of factors that can influence migration decisions.

Please also note the supplement to this comment:
https://esd.copernicus.org/preprints/esd-2019-79/esd-2019-79-AC3-supplement.pdf

---

## Author Comment (AC4) · 16 Jul 2020

Formatted version is attached as supplement.

Referee #2 was somewhat less enthusiastic than Referee #1. Referee #2 states:

The manuscript, without any doubt, studies an interesting and very important question in the academic and policy worlds. It also shows a methodological sophistication and rigor, . . .

However, the referee goes on to say:

Unfortunately, I believe that this manuscript does not advance our understanding of the climate-migration relationship and as a result, it does not make any significant

contribution to the relevant literature.

We are not sure what to say here. We have for the first time taken methods used by economists to estimate GDP changes and have applied them instead to predicting changes in population density, and using these predictions to infer possible incentives to migrate. We know of no other study that has attempted to quantify potential incentives to migrate quantitatively in this way. We are particularly surprised by this comment since the referee asks us to look at, because our treatment is much more nuanced than others cited positively by this referee. For example, Hsiang and Sobel (2016) assume examine a scenario in which people want to maintain the same global mean temperature, and use language like "tropical populations would have to travel distances greater than 1000 km" and "rapid evacuation of the tropics". We could have presented our study as a scenario and talked about all the people who would "have to migrate" under our scenario, but instead we chose to be more nuanced did not make any assumptions about people being tied to their current climates or whether they actually would migrate considering that there are many incentives not to move. We are somewhat perplexed in how to respond to a referee who cites a much simpler and less nuanced study to argue that we are not contributing anything new to the discussion. We are aware of no prior work that has performed a calculation like the one we are presenting in this work.

The referee writes:

However, at the end they chose to examine the climate-migration nexus in a direct and consequently deterministic manner, and make big and bold claims, which have huge implications of how the social consequences of climate change could be dealt with, based on false and naive assumptions.

Again, we are somewhat perplexed by the referee's remarks, as we are not aware of any big and bold claims that we are making and see nothing in our paper that could lead one to believe that we are treating "the climate-migration nexus in a direct and consequently deterministic manner".

The very first sentences of our abstract are:

Human migration is both motivated and constrained by a multitude of socioeconomic and environmental factors, including climate-related factors. Climatic factors exert an influence on local and regional population density. Here, we examine implications for future motivation for humans to migrate by analyzing today's relationships between climatic factors and population density, with all other factors held constant. Such 'all other factors held constant' analyses are unlikely to make quantitatively accurate predictions but the order-of-magnitude and spatial pattern that come out of such an analysis can be useful for thinking about the influence of climate change on the possible scale and pattern of future incentives to migrate.

Does this sound like the writing of people making big and bold claims "in a deterministic manner"?

Further, the very last paragraph of our manuscript reads:

Climate change is likely to induce a complex web of dynamical interactions at a range of spatial and temporal scales, and these interactions are not well represented by our model. For example, considerations of language, work, and family ties can provide strong incentive not to migrate. Projections of how climate change might affect migration are therefore fraught with uncertainty. Nevertheless, the results of our calculations may indicate areas that climate change can be expected provide large numbers of people, primarily in the tropics, an additional incentive to migrate, primarily to the middle and high latitudes of the Northern Hemisphere. This change in climate-driven incentives to migrate is one factor among many that need to be included in a comprehensive understanding of possible future migration flows.

Does this sound like the writing of people making big and bold claims "in a deterministic manner"? It is hard for us to understand what the referee is basing their comments on.

Usefully, the referee does point to some literature, which we now cite:

Black, Richard; W Neil Adger, Nigel W Arnell, Stefan Dercon, Andrew Geddes & David SG Thomas (2011) The effect of environmental change on human migration. Global Environmental Change 21: 3-11. Foresight Migration and Global Environmental Change. Final Project Report. London: The Government Office for Science, (2011). Available at: http://www.bis.gov.uk/foresight/ migration Hsiang, S.M. and A.H. Sobel (2016) Potentially extreme population displacement and concentration in the Tropics under non-extreme warming. Scientific Reports 6: 25697. McLeman, Robert & François Gemenne (2018) Environmental migration research: Evolution and current state of the science. In Robert McLeman & François Gemenne (ed.) Routledge Handbook of Environmental Displacement and Migration. London: Routledge, 3-16. Piguet, Etienne (2013) From "primitive migration" to "climate refugees": The curious fate of the natural environment in migration studies. Annals of the Association of American Geographers 103(1): 148-162. Rigaud, Kanta Kumari; Alex de Sherbinin, Bryan Jones, Jonas Bergmann, Viviane Clement, Kayly Ober, Jacob Schewe, Susana Adamo, Brent McCusker, Silke Heuser & Amelia Midgley (2018) Groundswell: Preparing for Internal Climate Migration, World Bank (https://openknowledge.worldbank.org/handle/10986/29461).

All of these references were easily incorporated into the existing text by adding an additional citation at the appropriate location. Because this is not a review paper, we did not attempt an exhaustive literature survey, so some sentences needed to be added to accommodate some of these suggested citations. The sentences we added are (to the Introduction):

Hsiang and Sobel (2016) examined consequences for migration if everyone moved to remain at the same annual global mean temperature under a climate change scenario.

And to the concluding paragraph:

A more complete treatment of migration, and not simply an examination of one possible set of incentives as we have done here, would require embedding our results in the

broader context of incentives that could influence migration decisions (Piguet, 2011)

And to the end of the Introduction:

When we report country-level results, we integrate across all grid cells within a country and report the net value, so our methodology would not predict incentive to migrate from a country that had some grid cells indicating incentives for out-migration but with other grid cells indicating even greater incentive for in-migration. Thus, internal migration is not considered in our study (Rigaud et al., 2018).

Please also note the supplement to this comment:
https://esd.copernicus.org/preprints/esd-2019-79/esd-2019-79-AC4-supplement.pdf